# The Triad Na^+^ Activated Na^+^ Channel (Nax)—Salt Inducible KINASE (SIK) and (Na^+^ + K^+^)-ATPase: Targeting the Villains to Treat Salt Resistant and Sensitive Hypertension

**DOI:** 10.3390/ijms24097887

**Published:** 2023-04-26

**Authors:** Sabrina R. Gonsalez, Dayene S. Gomes, Alessandro M. de Souza, Fernanda M. Ferrão, Zoe Vallotton, Venkateswara R. Gogulamudi, Jennifer Lowe, Dulce E. Casarini, Minolfa C. Prieto, Lucienne S. Lara

**Affiliations:** 1Faculdade de Medicina, Universidade Federal do Rio de Janeiro, Campus Macaé, Rio de Janeiro 21941-901, Brazil; srgonsalez@gmail.com; 2Instituto de Ciências Biomédicas, Universidade Federal do Rio de Janeiro, Rio de Janeiro 21941-971, Brazil; sgomesdayene@gmail.com (D.S.G.); alessandrocftq@gmail.com (A.M.d.S.); 3Núcleo Multidisciplinar de Pesquisa em Biologia (NUMPEX-BIO), Universidade Federal do Rio de Janeiro, Campus Caxias, Rio de Janeiro 21941-901, Brazil; fernandaferrao@xerem.ufrj.br; 4Department of Physiology, School of Medicine and Tulane Renal and Hypertension Center of Excellence, Tulane University School of Medicine, New Orleans, LA 70112, USA; zvallotton@tulane.edu (Z.V.); vgogulam@tulane.edu (V.R.G.); mprieto@tulane.edu (M.C.P.); 5Instituto de Biofísica Carlos Chagas Filho, Universidade Federal do Rio de Janeiro, Rio de Janeiro 21941-901, Brazil; lowe@biof.ufrj.br; 6Departamento de Medicina, Disciplina de Nefrologia, Escola Paulista de Medicina, Universidade Federal de São Paulo, São Paulo 04023-062, Brazil; casarini.elena@unifesp.br

**Keywords:** high-salt diet, DOCA salt rats, kidney function, chronic kidney disease, kidney fibrosis

## Abstract

The Na^+^-activated Na^+^ channel (Nax) and salt-inducible kinase (SIK) are stimulated by increases in local Na^+^ concentration, affecting (Na^+^ + K^+^)-ATPase activity. To test the hypothesis that the triad Nax/SIK/(Na^+^ + K^+^)-ATPase contributes to kidney injury and salt-sensitive hypertension (HTN), uninephrectomized male Wistar rats (200 g; *n* = 20) were randomly divided into 4 groups based on a salt diet (normal salt diet; NSD—0.5% NaCl—or high-salt diet; HSD—4% NaCl) and subcutaneous administration of saline (0.9% NaCl) or deoxycorticosterone acetate (DOCA, 8 mg/kg), as follows: Control (CTRL), CTRL-Salt, DOCA, and DOCA-Salt, respectively. After 28 days, the following were measured: kidney function, blood pressure, (Na^+^ + K^+^)-ATPase and SIK1 kidney activities, and Nax and SIK1 renal expression levels. SIK isoforms in kidneys of CTRL rats were present in the glomerulus and tubular epithelia; they were not altered by HSD and/or HTN. CTRL-Salt rats remained normotensive but presented slight kidney function decay. HSD rats displayed augmentation of the Nax/SIK/(Na^+^ + K^+^)-ATPase pathway. HTN, kidney injury, and kidney function decay were present in all DOCA rats; these were aggravated by HSD. DOCA rats presented unaltered (Na^+^ + K^+^)-ATPase activity, diminished total SIK activity, and augmented SIK1 and Nax content in the kidney cortex. DOCA-Salt rats expressed SIK1 activity and downregulation in (Na^+^ + K^+^)-ATPase activity in the kidney cortex despite augmented Nax content. The data of this study indicate that the (Na^+^ + K^+^)-ATPase activity response to SIK is attenuated in rats under HSD, independent of HTN, as a mechanism contributing to kidney injury and salt-sensitive HTN.

## 1. Introduction

Hypertension (HTN) is frequently aggravated by high salt (HS) consumption. However, the molecular mechanisms involved in the direct renal effects of elevated salt intake remain unclear. The adenosine monophosphate kinase (AMPK) family has been extensively studied due to the huge interest in cellular metabolism and its use as a pharmacological tool of metformin. The AMPK family holistically, though, has long overshadowed the role of its subfamily, the AMPK-related kinases; this sub-family includes salt-inducible kinase (SIK), which contains three isoforms (SIK1-3) [1,2,3]. SIK was initially identified in the adrenal gland of rats receiving a high-salt diet (HSD) [4]. All three SIK family members presented different molecular weights, but all shared conserved kinase ubiquitin-associated domains that were expressed broadly [5]. SIK1 mRNA expression is regulated by multiple stimuli, including (i) HSD intake, (ii) ACTH signaling [6], (iii) glucagon signaling [7], (iv) excitable cell depolarization [8], and (v) circadian rhythms [9]. SIK1 possesses several kinase regulatory sites, such as protein kinase A (PKA; T473 and S575), calmodulin-activated protein kinase 1 (CaMK1; Thr322), liver kinase B1 (LKB1; Thr182), and an autophosphorylation residue at Ser186 [5,10]. In humans, SIK2 and SIK3 are expressed ubiquitously, with the highest SIK2 and SIK3 levels in adipose tissue and in the brain, respectively. In a study by Bertorello et al. [11], it was shown that vascular SIK1 is involved in the development of HTN. In response to chronic salt intake (1% NaCl), vascular SIK1 functions to prevent high blood pressure [11]. 

SIK isoforms can function as intracellular translators of locally augmented extracellular Na^+^, but the upstream mechanism by which Na^+^ levels are sensed remains unknown. Previous studies have demonstrated how the atypical sodium channel, Nax (*scn7a*), acts as a Na^+^-sensing molecule in the central nervous system (CNS); it is critical to maintaining Na^+^ homeostasis in mammals [12,13]. Nax is a voltage-gated channel that has lost its characteristics and, instead, is a concentration-dependent sensor of extracellular Na^+^ and subsequent signal transducer [14,15]. We previously reported that rat kidney epithelial cells of the thick ascending limb (TAL) and principal cells (PC) of the collecting duct (CD) express Nax, which is upregulated by HSD. This finding suggests an essential role in monitoring changes in tubular fluid Na^+^ concentration [16].

In hypertension, salt sensitivity leads to renal hemodynamic and metabolic alterations and kidney injury, all of which aggravate the risk of cardiovascular and renal morbidity, and establish a vicious cycle. The renal tubular mechanisms of Na^+^ transport are driven by the primary regulators of extracellular fluid volume. The primary regulator of extracellular fluid volume is the basolateral (Na^+^ + K^+^)-ATPase in renal tubule epithelial cells. The mechanics of this specific Na^+^ transport is the limiting step of the Na^+^ reabsorption [17]. Sustained stimulation in the tubular Na^+^ reabsorption process is associated with the development of HTN.

In renal cell cultures, SIK1 regulates (Na^+^ + K^+^)-ATPase to its maximal capacity, by acting as a Na^+^ intracellular sensor that enables cells to amplify their Na^+^ signaling process [18,19,20,21]. Without this amplification, the Na^+^ signal might require a transient increase of a massive intracellular Na^+^ concentration that can lead to renal epithelial cell swelling. It has been shown that increased intracellular Na^+^ concentration, in response to monensin (a Na^+^ ionophore), leads to SIK1 activation. This activation occurs by a Ca^2+^-mediated signaling that activates subsequent inorganic pyrophosphatase 2 (PPA2), which contributes to (Na^+^ + K^+^)-ATPase dephosphorylation and its activation [20]. The correlation of HTN and SIK1 activation was demonstrated by using a mutant (hypertensive) form of α-adducin on (Na^+^ + K^+^)-ATPase activity [21]. In this study, the augmented (Na^+^ + K^+^)-ATPase activity elicited by the hypertensive variant of α-adducin was abolished by inhibiting the SIK1-dependent signaling. This event led to the hypothesis that increased SIK1 activity in the renal tubule epithelium may result in enhanced Na^+^ reabsorption that aids salt sensibility. It is unknown SIK isoforms localization in the kidney glomeruli and tubules and if the enzyme activity is modulated by HSD and/or HTN. Moreover, the α1 isoform of (Na^+^ + K^+^)-ATPase acts as a direct regulatory partner of the Nax channel. It directly influences Na^+^ influx by controlling the Na^+^ permeability of the Nax channel in the median preoptic nucleus of the third ventricle, which directly accesses the ionic composition of the cerebrospinal fluid [22].

In the present study, we aimed to investigate the role the triad Nax/SIK/(Na^+^ + K^+^)-ATPase plays in the development of salt-sensitive hypertension induced by deoxycorticosterone acetate (DOCA) leading to kidney injury. DOCA is an aldosterone precursor which increases Na^+^ and water reabsorption, which further promotes hypervolemia. The administration of HSD in combination with DOCA improves the onset of HTN [23]. Moreover, to determine the unique effect of the HS intake, we used uninephrectomized placebo-treated rats subjected to HSD or NSD. We propose that the overexpression of Nax and impaired regulation of SIK and (Na^+^ + K^+^)-ATPase in the kidney cortex are the molecular mechanisms contributing to kidney injury and exacerbation of HTN. 

## 2. Results

### 2.1. Physiological Parameters

At the end of 28 days, all rat groups reached the same body weight (Table 1). All three experimental conditions developed higher kidney indexes, with maximal development in the DOCA-Salt rats (45% increase in comparison to CTRL; *p* < 0.0001). As expected, the rats subjected to HSD presented an increased 24 h water intake (90 and 130% increase, CTRL-Salt and DOCA-Salt, respectively, in comparison to each corresponding control; *p* = 0.0173 and *p* = 0.0007) and increased 24 h urine volume (3 and 5 times, CTRL-Salt and DOCA-Salt, respectively, in comparison to each control; *p* = 0.0022 and *p* < 0.0001) (Table 1). HSD alone did not change systolic blood pressure (SBP), but the DOCA administration augmented SBP in ~40% of treated rats (*p* < 0.0001). The combination of HSD and DOCA administration exacerbated SBP (42% higher than CTRL-Salt, *p* < 0.0001; and 10% higher than DOCA, *p* = 0.0136). Heart rate (HR) increased across all experimental conditions. HR reached maximum levels in the DOCA-Salt rat (1.7 times higher than CTRL, *p* < 0.0001). HSD exacerbated the effects of HTN because the values of SBP, water intake, and urine volume in the DOCA-Salt were augmented in comparison to DOCA. HTN also intensified the effect of HSD since the values of SBP, HR, and kidney index in the DOCA-Salt were higher than the values observed in CTRL-Salt. The results demonstrated that DOCA-Salt rats developed salt-sensitive HTN. 

The unique impact of HSD intake was demonstrated in blood urea nitrogen (BUN) accumulation, proteinuria, and kidney-Na^+^-handling kidney function parameters. Compared to their respective controls, CTRL-Salt and DOCA-Salt showed ~1.3 times increased BUN and 2–3 times increased proteinuria (CTRL vs. CTRL-Salt, *p* = 0.0007; and DOCA vs. DOCA-Salt, *p* = 0.0497). Decreased kidney function was accentuated in the DOCA-Salt rat in comparison with both CTRL and DOCA. DOCA-Salt resulted in (i) a 30% increase in plasma creatinine (P_Cre_; *p* = 0.0043 and *p* = 0.0439); (ii) decreased urinary creatinine (U_Cre_, 80%; *p* < 0.0001 and *p* = 0.0037); and (iii) the P_Cre_/U_Cre_ ratio decreasing in 70% (*p* = 0.0002 and *p* = 0.0160) (Table 2). Plasma Na^+^ concentration did not vary in any of the experimental conditions. As expected, HSD provoked an intense urinary Na^+^ excretion (U_Na_): ~3 times in both CTRL-Salt and DOCA-Salt (*p* < 0.0001). DOCA-salt rats presented an augmented fractional Na^+^ excretion (FE_Na_) ~20 times (*p* < 0.0001), in addition to decay in 40% of filtered load Na^+^ (FL_Na_; *p* = 0.0243 and *p* = 0.0003). As a result, kidney function is jeopardized by the combination of HSD and HTN. HTN intensified the HSD effects in FL_Na_ and FE_Na_, but did not alter filtration kidney function. 

HTN—but not HSD—caused kidney tissue damage (Figure 1). DOCA rats presented a clear focus of mononuclear inflammation (arrow) and dilation of tubules with flattened epithelium which were light-filled with cast protein inside (asterisks) (compare Figure 1A,C). The same profile was observed in the DOCA-Salt rats, which was aggravated by the presence of microcalcification foci (arrowhead; Figure 1(D1)) and proximal tubules filled with red blood cells (filled circle) (Figure 1D). No renal morphological alterations were noted in CTRL (Figure 1A) and CTRL-salt (Figure 1B). Fibronectin accumulation was noticed in all three experimental conditions except CTRL; fibronectin deposition (arrowheads) was observed in the tubulointerstitum and the glomeruli (Figure 2A). Qualitative analysis indicated an increased intensity of fibronectin in CTRL-Salt, DOCA, and DOCA-Salt. 

### 2.2. SIK Isoform Localization along the Nephron

To assist in SIK kidney immunoexpressing cells and cell type identification, we used immunofluorescence histochemistry and the following markers: THY.1, (mesangial cell at glomeruli); aquaporin-1 (AQP1) (proximal tubule, PT); Tamm–Horsfall (TH) glycoprotein (thick ascending limb, TAL); and aquaporin-2 (AQP2) (principal cells of distal tubules and collecting duct) (Figure 3, Figure 4 and Figure 5). The colocalization results are summarized in Table 3. All three SIK isoforms were expressed along the nephron. Co-localization was marked in yellow in the merge images and noted as follows: (i) SIK1 and SIK3 are expressed in mesangial cells (Figure 3A and Figure 5A), (ii) SIK3 is expressed in the brush border of the proximal tubule cells (Figure 5B), and (iii) SIK3 is expressed in the principal cells (Figure 5D). SIK1 isoforms were not detected in the PT (Figure 3), and SIK2 was not detected in mesangial cells (Figure 4) or in the TAL (Figure 5). Immunolocalization did not change in any experimental conditions (Figure 6).

### 2.3. Influence of HSD and HTN on SIK Content and Activity in the Kidney Cortex

SIK1 protein content in the renal cortex was three-fold greater in CTRL-Salt and DOCA rats in comparison to the CTRL (*p* < 0.0001 and *p* = 0.0141; Figure 7). The combination of HSD and HTN downregulated SIK1 protein content to the CTRL levels [DOCA: 0.48 ± 0.05 a.u.; DOCA-Salt: 0.25 ± 0.03 a.u. (*p* = 0.0231); CTRL: 0.22 ± 0.04 a.u. (*p* = 0.9994)]. SIK1 protein content in DOCA-Salt was also diminished in comparison to CTRL-Salt (*p* < 0.0001). To determine SIK activity, we measured its ability to transfer phosphate from the [γ-P]ATP into the histone in the presence and absence of two SIK inhibitors: MRT 199665 (Figure 8A) and HG-9-91-01 (Figure 8B). HSD increased SIK activity in the kidney cortex regardless of inhibitor presence and specificity. The MRT 199665-sensitive SIK activity was 9.9 ± 2.4 in CTRL and 41.8 ± 4.2 pmol~P esterified × mg^−1^ in CTRL-Salt (*p* < 0.0001; Figure 8A). The HG-9-91-01-sensitive SIK activity was 10.5 ± 0.8 in CTRL and 22.7 ± 1.4 pmol~P esterified × mg^−1^ in CTRL-Salt (*p* < 0.0001; Figure 8B). MRT 199665-sensitive SIK activity was augmented in the DOCA-Salt rats (25.7 ± 1.5 pmol~P esterified × mg^−1^; *p* = 0.0163) but not in the DOCA rats (Figure 8A). However, the increase seen in the DOCA-Salt rats was lower than the levels observed in CTRL-Salt (*p* = 0.0142; Figure 8A). HG-9-91-01-sensitive SIK-activity was augmented exclusively in the CTRL-Salt rats (*p* < 0.0001; Figure 8B). HTN diminished HG-9-91-01-sensitive SIK activity in DOCA (6.2 ± 0.76 pmol~P esterified × mg^−1^) and in DOCA-Salt (6.3 ± 0.74 pmol~P esterified × mg^−1^; *p* = 0.0344 and *p* = 0.0399, respectively) in comparison to CTRL (Figure 8B). HG-9-91-01 sensitive SIK activity in DOCA-Salt was also diminished in comparison to CTRL-Salt (*p* < 0.0001; Figure 8B).

### 2.4. The Impact of SIK Activity in Kidney Cortex (Na^+^ + K^+^)-ATPase Activity

(Na^+^ + K^+^)-ATPase protein content in the kidney cortex was not altered in any of the experimental conditions (Figure 9A). However, HSD intake provoked a two-fold increase in (Na^+^ + K^+^)-ATPase activity (*p* = 0.0138; Figure 9B). As observed in the HG-9-91-01-sensitive SIK activity model, DOCA in combination with HSD decreased (Na^+^ + K^+^)-ATPase activity in comparison to CTRL and CTRL-Salt (*p* = 0.005 and *p* = 0.0008, respectively; Figure 9B). HTN alone did not change the transporter activity. 

To determine the effect of SIK on (Na^+^ + K^+^)-ATPase activity, we measured the Na^+^ transporter activity in the presence and absence of SIK inhibitor HG-9-91-01 (Figure 9C). In CTRL rats, 47% of the total (Na^+^ + K^+^)-ATPase activity was sensitive to SIK inhibition. In these rats, the total enzyme activity (the ATPase activity measured in the absence of HG-9-91-01) was 92.5 ± 6.6 nmol Pi × mg^−1^ × min^−1^, while in the presence of HG-9-91-01, the activity of (Na^+^ + K^+^)-ATPase was 41.7 ± 6.3 nmol Pi × mg^−1^ × min^−1^. The difference between (Na^+^ + K^+^)-ATPase activity measured in the absence and the presence of SIK inhibitors corresponds to the enzyme activity sensitive to SIK, which, in CTRL, was 50.8 ± 3.1 nmol Pi × mg^−1^ × min^−1^. In 73% of HSD rats, augmentation of the fraction of (Na^+^ + K^+^)-ATPase activity sensitive to HG-9-91-01 in comparison to CTRL was noted. In CTRL-Salt, the total enzyme activity was 169 ± 16 nmol Pi × mg^−1^ × min^−1^; in the presence of HG-9-91-01, it was 48.7 ± 16.4 nmol Pi × mg^−1^ × min^−1^; and the (Na^+^ + K^+^)-ATPase activity sensitive to SIK inhibition was 120 ± 25.7 nmol Pi × mg^−1^ × min^−1^ (*p* = 0.0137). In DOCA rats, the (Na^+^ + K^+^)-ATPase activity sensitive to SIK inhibition was almost undetected, and in DOCA-Salt, the SIK sensitivity was restored to CTRL levels (30.1 ± 2.9 nmol Pi × mg^−1^ × min^−1^ (*p* = 0.6893)—this was still decreased in comparison to CTRL-Salt (*p* = 0.0019). 

Another consequence of SIK inhibition due to HTN is TGFβ expression in the kidney. Figure 10B shows that TGFβ expression fainted in the kidney of CTRL-Salt rats. This effect was more accentuated in HTN induced by DOCA (Figure 10C,D). Quantification of TGFβ protein content in the kidney cortex demonstrated the same profile in comparison to CTRL: a 45% decrease in CTRL-Salt rats (*p* = 0.0046), a 50% decrease in DOCA rats (*p* = 0.0020), and a 60% decrease in DOCA-Salt rats (*p* = 0.0067). DOCA versus DOCA-Salt rats and CTRL-Salt versus DOCA-Salt rats presented similar levels of TGFβ expression.

Our study determined whether the protein content of the sodium sensor Nax was regulated under the examined experimental conditions (Figure 11). Nax is known for its transduction of increased extracellular Na^+^ concentration to intracellular signaling [22]. HSD intake induced a two-fold increase of Nax protein content in the kidney cortex (0.16 ± 0.02 CTRL-Salt vs. 0.086 ± 0.009 CTRL; *p* = 0.0379). DOCA also augmented Nax protein content in proportion to HSD intake (0.14 ± 0.009; *p* = 0.0479). Moreover, the combination of HSD and HTN promoted an exacerbated Nax expression similar to CTRL-Salt (0.22 ± 0.02 DOCA-Salt; *p* = 0.0189 and *p* = 0.1209, respectively; Figure 11).

## 3. Discussion

In this study, we demonstrated the immunolocalization of SIK isoforms in the kidney of normotensive rats under NSD and noted the following: (i) SIK1 is localized in mesangial cells, cells of the TAL, and principal cells of the collecting duct; (ii) SIK2 is localized in the glomerulus and throughout the tubular epithelial cells; and (iii) SIK3 is localized in mesangial cells, the DT, and the CD. Nax is localized in the same cell types as SIK1—cells of the TAL and CD [20]—indicating that those cells are crucial for translating augmented extracellular local Na^+^ concentration. HSD alone, HTN alone, and the combination of HSD and HTN showed no modification in the immunolocalization of SIK isoforms. HSD specifically increased Nax and SIK1 protein contents alongside increased (Na^+^ + K^+^)-ATPase activity. Although in rats kidney tissues presented normal structures, the resulting proteinuria, BUN accumulation, and decreased U_Cre_/P_Cre_ ratio indicate an alteration in kidney function. HTN induced by DOCA increased Nax and SIK1 protein contents, but the activities of SIK and (Na^+^ + K^+^)-ATPase are like those in CTRL rats. HTN caused kidney injury and a decreased U_Cre_/P_Cre_ ratio. On the other side, DOCA rats developed salt-sensitive HTN, as reflected by the increased values of SBP. The combination of HSD and HTN resulted in the worst scenario for the rats, characterized by higher levels of SBP, kidney injury, decay in renal function, and intense proteinuria. Impaired Na^+^ handling was also present as FL_Na_ was diminished and FE_Na_ was still higher than that in CTRL. The SIK1 protein content in the kidney cortex of DOCA-Salt rats was maintained to CTRL levels associated with a diminished (Na^+^ + K^+^)-ATPase activity and increased levels of Nax in comparison to CRTL rats, which may explain, in part, the pathogenic mechanism in salt-sensitive hypertension. The decreased (Na^+^ + K^+^)-ATPase activity is likely to be related to the higher level of FE_N_.

High salt consumption is a villain who works undercover. It works quietly, causing no detectable changes in kidney structure and function until a disturbance occurs. To demonstrate the effects of a diet with high content of Na^+^, Amara et al. [25] used an uninephrectomized rat model subjected to HSD. The rats presented impaired renal function but no significant modifications in cardiovascular parameters. The DOCA-Salt rat model is valuable and feasible for understanding the complexity of salt-sensitive HTN because it combines the effects of HSD and hypervolemia [26]. The crosstalk between neurogenic and endocrine components in the DOCA-Salt rat model contributes to severe HTN with kidney function impairment. Adding uninephrectomy to the model accelerates the onset and progression of HTN [27]. There is an interconnection between reduced renal function and glomerular damage, as indicated by the development of proteinuria and reduced creatinine clearance when DOCA administration is combined with HSD [26,27]. In the present study, the combination of HSD intake for four weeks and the DOCA administration resulted in a more severe renal dysfunction in DOCA-Salt rats, as evidenced by increased proteinuria, BUN, plasma creatinine, and decreased creatinine clearance. Moreover, alterations in kidney morphology, such as microcalcification foci and abundant red blood cells inside PT, contribute to the establishment of CKD, as previously described by other studies [24,26,27]. 

SIK plays a central role in the molecular regulation of BP during Na^+^ input [11]. In human genetic studies, a single nucleotide polymorphism resulting in one amino acid change (^15^Gly→Ser) in the protein enhances kinase activity and is associated with lower BP, suggesting the potential relevance of SIK1 in the development of HTN [28]. In the present study, HSD alone increases the percentage of SIK-sensitive (Na^+^ + K^+^)-ATPase activity. In contrast, DOCA-Salt rats (HTN in combination with HSD) exhibited the lowest levels of SIK1 in comparison to other experimental rats, which can be related to lower (Na^+^ + K^+^)-ATPase activity and a lower response to the SIK inhibitor HG 9-91-01. Decreased (Na^+^ + K^+^)-ATPase activity is likely to lead to a diminished Na^+^ reabsorption, likely due to an adaptive mechanism, since augmented Na^+^ reabsorption is related to the development of HTN in many studies using animals and humans with HTN [29,30]. We can speculate that this mechanism may lead to kidney injury and salt sensitivity. This effect in the kidney is the opposite of that observed in vascular smooth muscle cells. The overexpression of the rare allele (SIK1-^15^Ser) increases (Na^+^ + K^+^)-ATPase activity. This event is associated with a decrease in intracellular Ca^2+^ concentration, which reduces vascular tone [31,32].

The immunolocalization of SIK revealed that PT does not express SIK1, as reported elsewhere. Most data use tubular epithelial cell cultures like opossum kidney cells (OK cells) [20], Madin–Darby canine kidney cells (MDCK cells), primary cultures of rat PT cells (RPT) [18], and renal proximal tubule cells from Milan hypertensive rats (carrying a α-adducin mutation) [21]. The fact that the same antibody we used to immunolocalize SIK isoforms in the renal cortex gave a positive result (Appendix A) may suggest reduced specificity of the antibody and the possibility of binding other proteins in cultured PT cells.

SIK isoforms are expressed in the epithelial cells of the TAL and CD. Equally important, Nax is expressed on the luminal side of the same tubules [16]. The TAL is the first segment of the distal nephron, extending from cortex to outer medulla. These two regions differ in structural and functional characteristics. Superficial nephrons present a long cortical TAL that runs in the medullary rays neighboring the pars recta of the PT and cortical CD. The abundant network of capillaries issued from the efferent arterioles of superficial and mid-glomeruli establishes itself as an adequate environment to sense local increases in Na^+^ concentration [33]. Moreover, the CD is involved in Na^+^ homeostasis. Na^+^ influx through the epithelial Na^+^ channels (ENaC) induces a coordinated increase in basolateral (Na^+^ + K^+^)-ATPase expression and activity. Signaling through kinases plays a key role in this process and may rely on the modulation of a Na^+^ sensor that remains to be identified [34].

The presence of Nax and SIK isoforms in the epithelial cells of the TAL and the CD indicates their involvement in the molecular mechanism of vectorial Na^+^ reabsorption activated by the (Na^+^ + K^+^)-ATPase. Humans and animal models with HTN exhibit increased kidney Na^+^ reabsorption, which further supports the relevance of the nephron Na^+^ signaling pathway in the development of HTN [17,35]. SIK1 regulates the maximal capacity of (Na^+^ + K^+^)-ATPase, acting as a Na^+^ intracellular sensor, enabling renal cells to amplify its Na^+^ signaling process [19]. Cooperative actions between the Nax channel and the α1 isoform of (Na^+^ + K^+^)-ATPase generate a functional active complex in the median preoptic nucleus of the third ventricle. This plays a central role in the control of Na^+^ influx under physiological conditions [22]. This observation can be extrapolated to the kidneys as HSD alone increases (Na^+^ + K^+^)-ATPase activity.

The triad Nax/SIK/(Na^+^ + K^+^)-ATPase is regulated by both HSD and HTN, which results in a dual effect (Figure 12). HSD overexpresses and overstimulates the triad. The high blood pressure also overexpresses the triad, but SIK and (Na^+^ + K^+^)-ATPase activities are close to control levels. Downregulation of the triad is evidenced in the combination of HSD and HTN. Although DOCA-Salt rats elicited the highest levels of Nax, both enzymes presented levels of expression and activity close to that of CTRL rats. 

Yet an intriguing question remains. How do TGF-β protein levels and immunoexpression decrease in all three experimental groups, as TGF-β is an inflammatory cytokine associated with renal fibrosis in several progressive kidney diseases [35,36]? SIK negatively regulates TGF-β signaling in a model of HaCaT cells. When HaCaT cells are stimulated with TGF-β after endogenous SIK depletion, the cells show an enhanced TGF-β signaling. This was reported by measuring mRNA levels of well-characterized TGF-β gene targets [37]. The kidneys of CTRL-Salt rats showed decreases in SIK1 protein expression and activity in the renal cortex and TGF-β downregulation. Thus, we postulated that the downregulation of TGF-β might be an adaptive mechanism in response to HSD. In HTN, TGF-β1 is also downregulated independent of HSD intake. Nax is also associated with kidney fibrosis. Both the gene and the protein expressions of Nax were upregulated in several models of murine and human kidney fibrosis [14,38]. In the present study, DOCA-Salt rats elicited intense Nax expression in the renal cortex, which may contribute to increased fibronectin deposition and may help to explain the aggravated renal injury. Indeed, exacerbation of homeostatic Na^+^ regulation via Nax proceeds to induce pathological tissue phenotypes via the promotion of pro-inflammatory and pro-fibrotic responses [14]. Further studies are required to clarify the underlying mechanisms involved in these responses.

Syndemic is an emergent concept that characterizes the mutually aggravating interactions between two or more concurrent or sequential epidemics or disease clusters in populations associated with socioeconomic status and environmental factors [39]. It has been well established that syndemic conditions of low education and poor quality of population nutrition—childhood undernourishment, impaired mental health, viral infection (COVID-19 and HIV), drug abuse, and obesity—are risk factors for the silent development of cardiovascular diseases, such as HTN [40]. HTN affects 1.28 billion people worldwide and is considered a significant risk factor for heart disease, stroke, and kidney failure, resulting in premature death and disability [41,42]. To date, most literature on syndemic HTN involves HSD and kidney disease. Uninephrectomized Wistar rats subjected to HSD proves to be a salt-resistant model because it lacks an HTN phenotype and allows detection of the unique effect of Na^+^ intake on the kidney (Figure 12). The evidence of normal kidney tissue structure, proteinuria presence, and a decreased U_Cre_/P_Cre_ ratio is compatible with the initial phase of CKD. The association of HSD with DOCA promotes salt-sensitive HTN, which speeds up the impairment of kidney function, as reflected by the decay of renal physiological parameters. At this point, chronic kidney disease is established. The presence of the high BP component negatively regulates SIK activity, which promotes dysregulation of (Na^+^ + K^+^)-ATPase. This observation may explain the augmentation in FENa despite the low levels of FLNa in DOCA-Salt rats. While DOCA-Salt rats do not perfectly mimic the complexity of CKD in its entirety, CKD was evidenced by the onset of renal injury and HTN. Both were exacerbated in association with HSD. Thus, we postulate that the triad Nax/SIK/(Na^+^ + K^+^)-ATPase is the molecular key to preventing and treating the consequences of salt-sensitive HTN. 

## 4. Materials and Methods

### 4.1. Experimental Design

Twenty 8-week-old male Wistar rats weighing 200 g each were purchased from Centro de Criação de Animais de Laboratório (CECAL, Fundação Oswaldo Cruz, Rio de Janeiro, Brazil). The rats were maintained in the vivarium under the following conditions: (i) constant temperature (23 ± 2 °C), (ii) standard dark/light cycles (12/12 h), and (iii) chow and water distribution *ad libitum*, as recommended by good practice standards in research and approved by the Ethics Committee for the Use of Animals of the Universidade Federal do Rio de Janeiro, Brazil (protocol number 138/13). On day 0, the rats underwent unilateral nephrectomy of the right kidney, as previously described by Lindoso et al. [24]. After recovery, the rats were randomly divided into 4 groups: (i) control (CTRL): rats were fed a normal salt diet (NSD, 0.5% NaCl) plus subcutaneous administration of saline (0.9% NaCl); (ii) high-salt diet (CTRL-Salt): rats were fed an HSD (containing 4% NaCl) plus subcutaneous administration of saline (0.9% NaCl); (iii) DOCA: rats were fed an NSD plus subcutaneous administration of 8 mg/kg deoxycorticosterone acetate (DOCA) twice a week; (iv) DOCA-Salt: rats were fed an HSD (containing 4% NaCl) plus subcutaneous administration of 8 mg/kg DOCA twice a week. Diet-specific food given to the rats was purchased from Rhoster Industria e Comércio Ltd. (Araçoiaba da Serra, São Paulo, Brazil). After 28 days of treatment, rats were individually housed in metabolic cages to track water intake and urine excretion for 24 h before being euthanized. The remaining (left) kidney from each rat was removed and the cortexes were sectioned for protein quantification as described in [43] and total membrane preparation was obtained as previously described by Amara et al. [25]. Immediately after kidney harvesting, the whole left kidney of each rat was immersed in 4% paraformaldehyde for histology and immunofluorescence studies.

### 4.2. Non-Invasive Systolic Blood Pressure (SBP) and Heart Rate (HR) Measurements

At the beginning of the study (prior to rat randomization) and on day 28, SBP and HR were measured by tail-cuff plethysmography (digital pressure meter, LE 5000, Letica SA, Barcelona, Spain). Rats were kept in the equipment at a constant temperature of 30 ± 2 °C for 30 min until tail pulses were detected. After pulse stabilization, SBP and HR were measured three consecutive times consecutively for each animal. The rats were exposed to the entire procedure one to two days prior to taking the data to acclimatize and avoid rat stress [44]. 

### 4.3. Urine and Blood Analysis

The blood samples were collected after euthanasia in glass tubes containing 5 mM EDTA and centrifuged at 3000× *g* for 10 min to allow for the separation of plasma fractionation. Blood urea nitrogen (BUN), Na^+^, and creatinine were measured in plasma samples. Urine samples were centrifuged for 5 min to eliminate sediments prior to proteinuria, Na^+^, and creatinine analysis. Urinary and plasma creatinine, BUN, and proteinuria were measured by spectrophotometry using specific colorimetric kits (Gold Analisa, Belo Horizonte, Brazil). Renal function parameters were calculated as previously described by Cortes et al. [45]. 

### 4.4. Renal Histology and SIK Immunolocalization

Kidney sections (4 μm) obtained from paraffin-embedded blocks were processed and stained with Hematoxylin-Eosin (HE) as described [45,46]. Immunofluorescence on kidney sections was carried out as described in previous studies [16,46,47]. Fluorescence amplification by tyramide signal amplification (TSA)-conjugated fluorophores was used to detect various targets in the same assay. The kidney sections were incubated with SIK 1, 2, and 3 antibodies (1:20 anti-rabbit polyclonal HPA038211, Sigma-Aldrich, St. Louis, MO, USA; 1:50 anti-rabbit NBP1-76572, Novusbio; 1:100 anti-chicken NBF241147, Novusbio, Centennial, CO, USA, respectively) overnight at 4 °C, followed by 1 h of incubation at 22 °C with the secondary antibody. Colocalization samples were placed by sequential incubation at room temperature for another 30 min with blocking serum, followed by 1 h with different immunomarkers: (1) anti-rat aquaporin-1 antibody (AQP1; 1:1000; ab9566; Abcam, Cambridge, UK), a marker of proximal tubule cells; (2) anti-rat aquaporin-2 antibody (AQP2; 1:100; sc-9982; Santa Cruz Biotechnology), a marker of the principal cells of connecting tubules and collecting ducts; (3) anti-rat TH antibody (Tamm–Horsfall; 1:500; sc-217023; Santa Cruz Biotechnology), a marker for TAL cells; and (4) anti-rat THY.1 antibody (1:500; ab225; Abcam, Cambridge, UK), a marker for mesangial cells. The samples were washed with TBS-T before applying donkey anti-rabbit HRP (ab6802; Abcam, Cambridge, UK) and Alexa Fluor 647 to both—this resulted in red fluorescence (Invitrogen, Waltham, MA, USA). After two simultaneous incubations of the first antibodies and three TBS-T washes, the sections were incubated with two simultaneously acting secondary antibodies: donkey anti-rabbit HRP (ab6802; Abcam, Cambridge, UK)—TSA substrate and Alexa Fluor 647 (red)—followed by incubation TSA (green fluorescein—Perkin Elmer FP1168, Waltham, MA, USA) diluted (1:200) in amplification. After five TBS-T washes of 5 min each, each slide was placed on the microscope for digital images captured using three fields from three different sets of cell cultures using 40× magnification (Nikon Eclipse 50i fluorescence microscope).

### 4.5. Western Blot Analysis

Membrane fractions from the kidney cortexes (80 µg of total protein) were separated by electrophoresis in a NOVEX 8% bis-Tris precast gel (Invitrogen system), transferred to a nitrocellulose NOVEX system, and then the membrane was incubated for 1 h with block solution. Membranes were incubated overnight at 4 °C with corresponding primary antibodies: mouse TGF-β1 (1:200; sc-52893; Santa Cruz Biotechnology); Nax rabbit Scn7a polyclonal antibody (1:500; Abcam, Cambridge, UK); rabbit SIK1 polyclonal antibody (1:500; sc-83754; Santa Cruz Biotechnology, Dallas, TX, USA); and rabbit (Na^+^ + K^+^)-ATPase polyclonal antibody (1:1000; 06-520; Merck Millipore). Next, the membranes were washed 3 times for 5 min with TBS. They were then incubated at room temperature with the secondary fluorescent antibodies, which were 10 times more diluted than the respective primary antibody (anti-rabbit, Li-Cor, IRDye 800CW). The resulting immunofluorescence was detected using the Odyssey System (Li-Cor Bioscience, Lincoln, NE, USA) for infrared imaging recording. Quantification analysis was done using Image J software (https://imagej.nih.gov/ij/download.html). β-actin immunostaining was used as a loading control (β-actin antibody 1:5000; A5316; Sigma-Aldrich).

### 4.6. SIK Activity

SIK activity was measured by the phosphorylation of histone (phospho-histone), which was used due to its sensitivity to its inhibitors (MRT199665 or HG-9-91-01; kindly donated by Professor Sir Philip Cohen , University of Dundee, UK), adapted from previous studies [44,48]. The authors sincerely thank, for kindly providing the SIK inhibitors MRT199665 and HG-9-91-01. Renal cortex homogenates (0.1 mg/mL) were briefly incubated for 10 min in the presence and absence of SIK inhibitors, respectively: 1 μM MRT199665 (inhibits SIKs and other members of the AMPK subfamily of protein kinases) and 1 μM HG-9-91-01 (inhibits SIKs and several protein tyrosine kinases but does not inhibit any other AMPK family member). Histone (0.2 mg/mL) was added in a reaction solution containing MgCl_2_ 4 mM, HEPES-Tris (pH 7.4) 20 mM, NaF 12 mM, ATPNa_2_ 10 µM, and the previously incubated homogenates, with or without SIK inhibitors, at 37 °C. The reaction began by adding [γ-^32^P]ATP (10 μmol/L; specific activity ~ 1.5 × 10^11^ Bq/mmol) to the renal homogenate sections. After 10 min, the reaction was stopped by adding 0.1 mL 40% (*w*/*v*) TCA; the samples were immediately placed on ice. The samples were vigorously stirred, and a 0.1 mL aliquot was removed, filtered through a Millipore filter (0.45 μm pore size), and successively washed with ice-cold 20% (*w*/*v*) TCA and 0.1 mol/L phosphate buffer (pH 7.0). Radioactivity was quantified by liquid scintillation. SIK-mediated phosphorylation was quantified by the difference between the levels of phospho-histone per mg of kidney cortex tissue in the absence and presence of the corresponding inhibitor.

### 4.7. Activity of Sodium Transporter

The (Na^+^ + K^+^)-ATPase activity was measured as described previously [17]. To examine the SIK-related impact on sodium transport, the activity of the transporter was measured in the presence and absence of 1 μM HG-9-91-01 SIK inhibitor. The enzyme sensitivity to SIK inhibition was calculated by the ATPase activity in the absence (total activity) and presence of HG-9-91-01 (activity resistant to SIK inhibition).

### 4.8. Statistical Analysis

The data are presented as the mean ± SEM. A one-way ANOVA followed by Sidak’s post hoc test was applied to detect differences in the effects of the following: (i) HSD alone (CTRL vs. CTRL-Salt rats, represented by letter “a” for *p* < 0.05); (ii) HTN alone (CTRL vs. DOCA, represented by letter “b” for *p* < 0.05); (iii) the combination of HSD and HTN (CTRL vs. DOCA-Salt rats, represented by letter “c” for *p* < 0.05); (iv) to see if HSD exacerbates the HTN effect (DOCA vs. DOCA-Salt rats, represented by letter “d” for *p* < 0.05); and (v) to see if HTN exacerbates the HSD effect (CTRL-Salt vs. DOCA-Salt rats, represented by letter “e” for *p* < 0.05). Statistical tests and graphs used were generated using GraphPad Prism 9.4 software (GraphPad Inc., La Jolla, CA, USA). 

## 5. Conclusions

SIK isoforms localize throughout the nephron: (i) SIK1 is present in mesangial cells, the thick ascending limb and the collecting duct; (ii) SIK2 is present in glomeruli, the proximal tubule, the thick ascending limb, and the collecting duct; and (iii) SIK3 is present in mesangial cells, the proximal tubule, the thick ascending limb, and principal cells of the collecting duct.Effect of HSD (CTRL-Salt vs. CTRL rats): CTRL-Salt rats elicit slight kidney function alteration, but not hypertension. In addition, these rats have augmented Nax content, total SIK activity, SIK1 content, and (Na^+^ + K^+^)-ATPase activity that is sensitive to SIK inhibition.Effect of HTN (DOCA vs. CTRL rats): DOCA rats exhibit moderate kidney function alteration and kidney injury along with augmented Nax and SIK1 contents, decreased total SIK activity, and unaltered (Na^+^ + K^+^)-ATPase activity that was insensitive to SIK inhibition.Effect of the combination of HSD and HTN:DOCA-Salt vs. DOCA rats: DOCA-Salt rats display exacerbation in SBP, severe kidney function decay, and kidney injury. In addition, these rats exhibit the greatest augmentation of Nax and SIK1 content, unaltered total SIK activity, and decreased (Na^+^ + K^+^)-ATPase activity sensitive to SIK inhibition.DOCA-Salt vs. CTRL-Salt rats: DOCA-Salt rats presented augmented SBP, HR, and a jeopardized tubular Na^+^ function, as FENa was decreased. This may be in part due to the diminished levels of SIK1, SIK activity, (Na^+^ + K^+^)-ATPase activity, and sensibility to SIK. However, compared to CTRL, Nax levels were increased in DOCA-Salt, which may contribute to the pathogenesis of salt sensitivity in this model. Fibrosis was attributed to augmented Nax content in CTRL-Salt, DOCA, and DOCA-Salt rats.We have identified the triad Nax/SIK/(Na^+^ + K^+^)-ATPase as the molecular key to preventing or treating the consequences of salt-sensitive HTN. HSD itself activates the triad Nax/SIK/(Na^+^ + K^+^)-ATPase, but on its own does not lead to HTN, although it initiates the progression of kidney function decay. In the present study, we demonstrated a disruption of the triad in the salt-sensitive HTN rats (DOCA-Salt rats). This is characterized by the intense expression of Nax and the attenuated response of SIK/(Na^+^ + K^+^)-ATPase in comparison to DOCA and CTRL-Salt rats. Thus, we conclude that Nax is associated with increased renal Na+ excretion and tissue fibrosis, which ultimately exacerbate HTN and the progression of chronic kidney disease. 

## Figures and Tables

**Figure 1 ijms-24-07887-f001:**
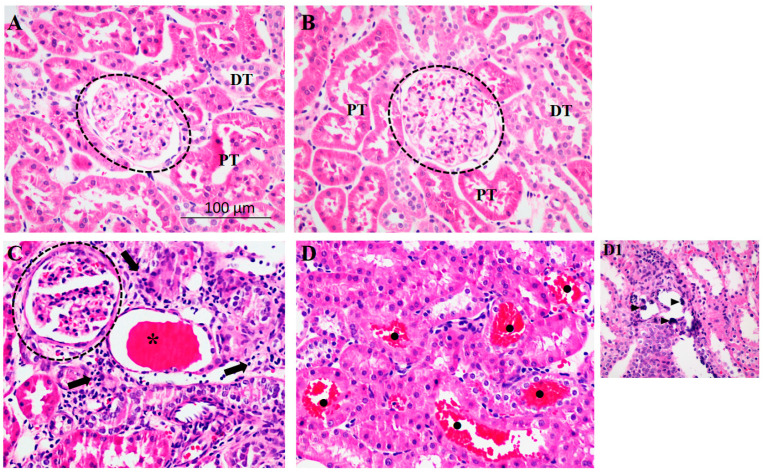
High-salt diet (HSD) associated with hypertension (HTN) exacerbates kidney injury. Representative photomicrographs (×400) of kidney sections (4 μm) stained with hematoxylin-eosin from uninephrectomized rats subjected to (**A**) normal salt diet (NSD; CTRL), (**B**) HSD (CTRL-Salt), (**C**) NSD and treated with deoxycorticosterone acetate (DOCA), and (**D**) HSD treated with DOCA (DOCA-Salt). PT: proximal tubule; DT: distal tubule. Arrows: mononuclear inflammatory infiltration. Asterisks: dilated tubule with flattened epithelium and light-filled with cast proteins inside. Closed circles: proximal tubules filled with red blood cells. (**D1**): Inset to Figure 1D. Arrowhead: microcalcification foci.

**Figure 2 ijms-24-07887-f002:**
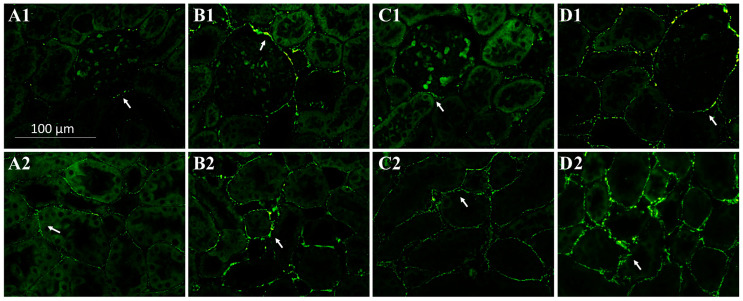
High-salt diet (HSD) and/or hypertension (HTN) contributes to fibronectin deposition in kidney tissue. Immunofluorescence was performed as described in Materials and Methods. Paraffin embedded kidney outer (1 series) and inner (2 series) cortex from CTRL (**A**), CTRL-Salt (**B**), DOCA (**C**), and DOCA-Salt (**D**) were incubated overnight with anti-rabbit fibronectin antibody (1:250) at 4 °C, followed by incubation with specific secondary antibody (1:1000, Alexa Fluor 488-labeled secondary antibody, Invitrogen, Waltham, MA, USA). Representative photomicrographs (×400) of kidney poles in glomeruli (upper panels) and tubulointerstitial areas (lower panels). Arrows demonstrate fibronectin deposition.

**Figure 3 ijms-24-07887-f003:**
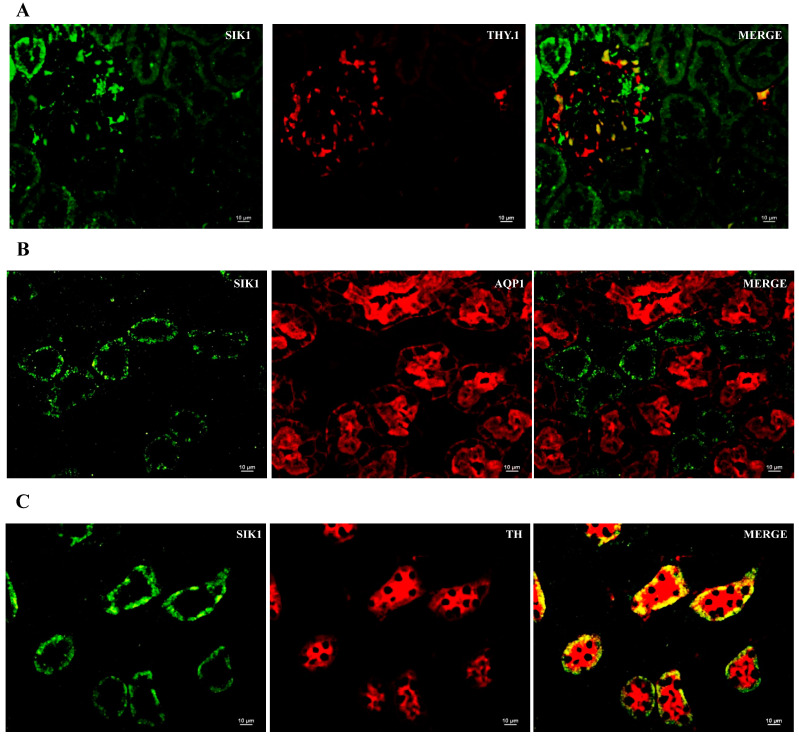
Immunolocalization of the salt-inducible kinase 1 (SIK1) in the normal rat kidney. Paraffin-embedded rat kidney sections were incubated with SIK1 rabbit polyclonal HPA038211 (1:20, Sigma-Aldrich, St. Louis, MO, USA) overnight at 4 °C, followed by incubation with green anti-rabbit secondary antibody (1:2000, Alexa Fluor 488, Invitrogen, Waltham, MA, USA). Representative photomicrographs of the kidney cortex. Immunolocalization of the SIK1 was addressed using specific immunomarkers: (**A**) THY.1 antibody for the mesangial cells 1:500, Abcam, Cambridge, UK); (**B**) anti-rat aquaporin—1 (AQP1) for the proximal tubule (1:100, Abcam, Cambridge, UK); (**C**) Tamm–Horsfall (TH, 1:500, Santa Cruz Biotechnology, Dallas, TX, USA) for the thick ascending limb; (**D**) anti-rat aquaporin—2 for principal cells of the collecting duct (AQP2; 1:100, Santa Cruz Biotechnology, Dallas, TX, USA).

**Figure 4 ijms-24-07887-f004:**
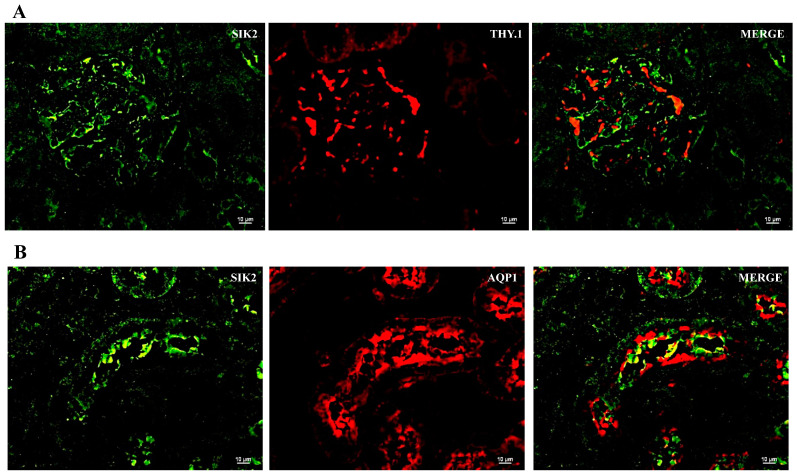
Immunolocalization of the salt-inducible kinase 2 (SIK2) in the normal rat kidney. Paraffin-embedded rat kidney sections were incubated with SIK2 rabbit polyclonal anti-Scn7a (1:100, Abcam, Cambridge, UK) overnight at 4 °C, followed by incubation with green anti-rabbit secondary antibody (1:2000, Alexa Fluor 488, Invitrogen, Waltham, MA, USA). Representative photomicrographs of the kidney cortex. Immunolocalization of the SIK2 was addressed using specific immunomarkers: (**A**) THY.1 antibody for the mesangial cells 1:500, Abcam, Cambridge, UK); (**B**) anti-rat aquaporin—1 (AQP1) for the proximal tubule (1:100, Abcam, Cambridge, UK); (**C**) Tamm–Horsfall (TH, 1:500, Santa Cruz Biotechnology, Dallas, TX, USA) for the thick ascending limb; (**D**) anti-rat aquaporin—2 (AQP2; 1:100, Santa Cruz Biotechnology, Dallas, TX, USA).

**Figure 5 ijms-24-07887-f005:**
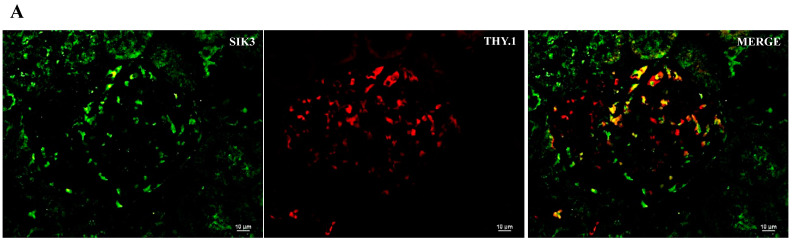
Immunolocalization of the salt-inducible kinase 3 (SIK3) in the normal rat kidney. Paraffin-embedded rat kidney sections were incubated with SIK3 rabbit polyclonal anti-Scn7a (1:100, Abcam, Cambridge, UK) overnight at 4 °C, followed by incubation with green anti-rabbit secondary antibody (1:2000, Alexa Fluor 488, Invitrogen, Waltham, MA, USA). Representative photomicrographs of the kidney cortex. Immunolocalization of the SIK3 was addressed using specific immunomarkers: (**A**) THY.1 antibody for the mesangial cells 1:500, Abcam, Cambridge, UK); (**B**) anti-rat aquaporin—1 (AQP1) for the proximal tubule (1:100, Abcam, Cambridge, UK); (**C**) Tamm–Horsfall (TH, 1:500, Santa Cruz Biotechnology, Cambridge, UK) for the thick ascending limb; (**D**) anti-rat aquaporin—2 (AQP2; 1:100, Santa Cruz Biotechnology, Cambridge, UK).

**Figure 6 ijms-24-07887-f006:**
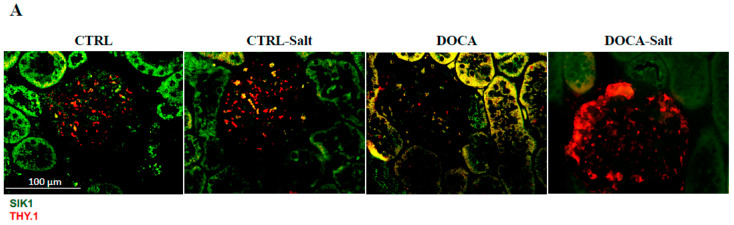
Immunolocalization of the salt-inducible kinase 1 (SIK1) was not altered by high-salt diet (HSD) or hypertension (HTN). Paraffin-embedded rat kidney from CTRL, CTRL-Salt, DOCA, and DOCA-Salt sections were incubated with SIK 1 rabbit polyclonal HPA038211 (1:20, Sigma-Aldrich, St. Louis, MO, USA) overnight at 4 °C, followed by incubation with green anti-rabbit secondary antibody (1:2000, Alexa Fluor 488, Invitrogen, Waltham, MA, USA). Representative photomicrographs of the kidney cortex. Immunolocalization of the SIK1 was addressed using specific immunomarkers: (**A**) THY.1 antibody for the mesangial cells (1:500, Abcam, Cambridge, UK); (**B**) anti-rat aquaporin—1 (AQP1) for the proximal tubule (1:100, Abcam, Cambridge, UK); (**C**) Tamm–Horsfall (TH, 1:500, Santa Cruz Biotechnology, Dallas, TX, USA) for the thick ascending limb; (**D**) anti-rat aquaporin—2 for principal cells of the collecting duct (AQP2; 1:100, Santa Cruz Biotechnology, Dallas, TX, USA).

**Figure 7 ijms-24-07887-f007:**
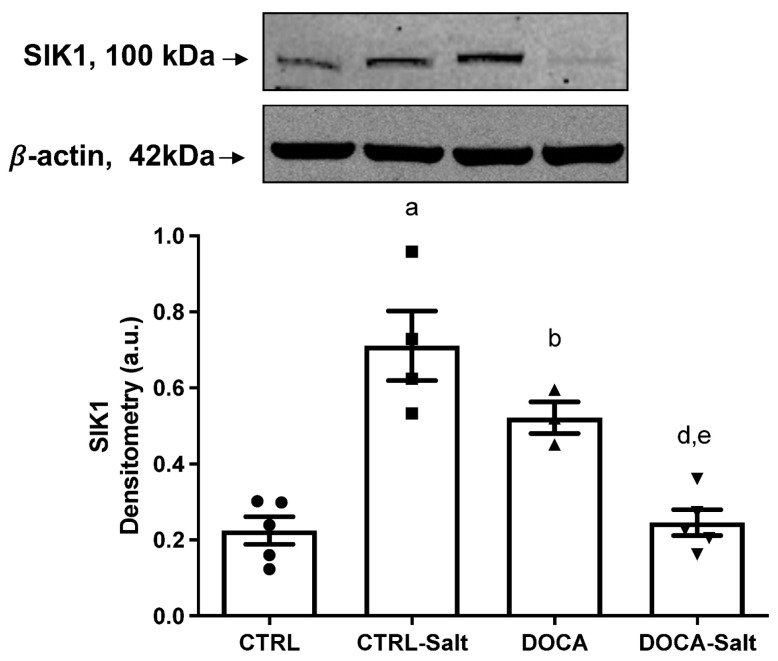
Effect of high-salt diet (HSD) in combination or not with hypertension (HTN) on SIK1 protein content in the kidney cortex from rats subjected to (i) normal salt diet (NSD; CTRL), (ii) HSD (CTRL-Salt), (iii) NSD and treated with deoxycorticosterone acetate (DOCA), and (iv) HSD treated with DOCA (DOCA-Salt). Upper panel: representative images of SIK1 detection using rabbit polyclonal HPA038211 (1:200, Sigma-Aldrich, St. Louis, MO, USA) and polyclonal β-actin (1:5000, Santa Cruz Biotechnology, Dallas, TX, USA). Lower panel: densitometric analysis of the immunoreactive band (100 kDa) correlated by the β-actin expression. The data are presented as the mean ± SEM. Different superscript lowercase letters indicate statistically significant differences (*p* < 0.05). One-way ANOVA followed by Sidak’s post hoc test was applied to detect differences in the effects of HSD (^a^—CTRL vs. CTRL-Salt); HTN (^b^—CTRL vs. DOCA); the combination of HSD and HTN (^c^—CTRL vs. DOCA-Salt); HSD-exacerbated HTN (^d^—DOCA vs. DOCA-Salt); and HTN-exacerbated HSD (^e^—CRTL-Salt vs. DOCA-Salt). Different superscripted lowercase letters represented in the figure only in case of statistically significant differences (*p* < 0.05).

**Figure 8 ijms-24-07887-f008:**
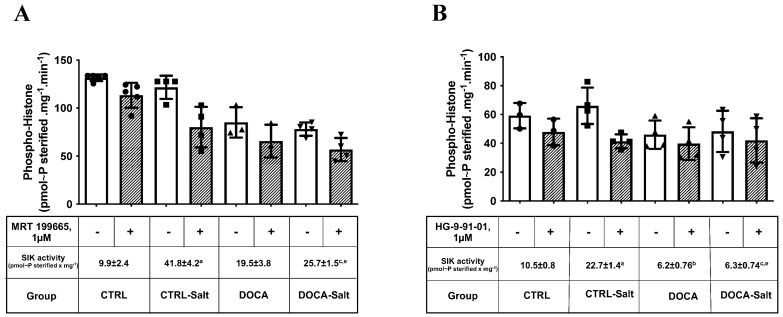
Effect of high-salt diet (HSD) in combination or not with hypertension (HTN) on salt-inducible kinase (SIK) activity in the kidney cortex from rats subjected to (i) normal salt diet (NSD; CTRL), (ii) HSD (CTRL-Salt), (iii) NSD and treated with deoxycorticosterone acetate (DOCA), and (iv) HSD treated with DOCA (DOCA-Salt). SIK activity was determined by the phosphorylation of histone (phospho-histone) sensitive to its inhibitor: (**A**) 1 µM MRT 199665 or (**B**) 1 µM HG-9-91-01, as described in Materials and Methods. The difference of histone phosphorylated in the absence (open bars) and in the presence (hachured bars) of the respective inhibitor represents SIK activity (values indicated in the tables below the graphs). The data are presented as the mean ± SEM. Different superscript lowercase letters indicate statistically significant differences (*p* < 0.05). One-way ANOVA followed by Sidak’s post hoc test was applied to detect differences in the effects of HSD (^a^—CTRL vs. CTRL-Salt); HTN (^b^—CTRL vs. DOCA); the combination of HSD and HTN (^c^—CTRL vs. DOCA-Salt); HSD-exacerbated HTN (^d^—DOCA vs. DOCA-Salt); and HTN-exacerbated HSD (^e^—CRTL-Salt vs. DOCA-Salt). Different superscripted lowercase letters represented in the Figure only in case of statistically significant differences (*p* < 0.05).

**Figure 9 ijms-24-07887-f009:**
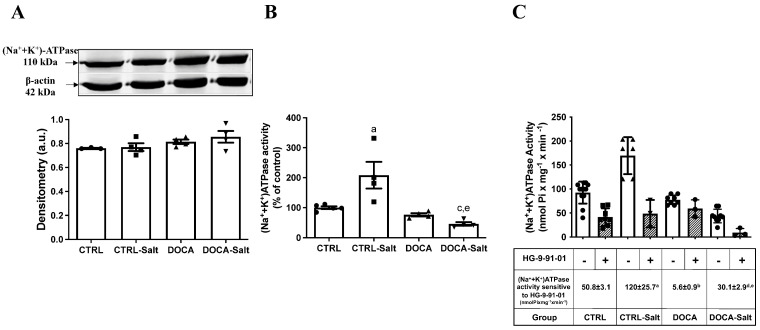
Effect of high-salt diet (HSD) in combination or not with hypertension (HTN) on (Na^+^ + K^+^)-ATPase content and activity and the impact of SIK inhibition. Wistar rats were subjected to (i) normal salt diet (NSD; CTRL), (ii) HSD (CTRL-Salt), (iii) NSD and treated with deoxycorticosterone acetate (DOCA), and (iv) HSD treated with DOCA (DOCA-Salt), as described in Materials and Methods. (**A**) (Na^+^ + K^+^)-ATPase content in kidney cortexes. Upper panel: representative images of (Na^+^ + K^+^)-ATPase detection using rabbit polyclonal anti-Na^+^/K^+^ ATPase α-1 (1:1000, Merck Millipore, Burlington, MA, USA) and polyclonal β-actin (1:5000, Santa Cruz Biotechnology, Dallas, TX, USA). Lower panel: densitometric analysis of the immunoreactive band (110 kDa) correlated by the β-actin expression. (**B**) (Na^+^ + K^+^)-ATPase activity. Ouabain-sensitive ATPase activity was measured in the homogenate obtained from the kidney cortex. (**C**) (Na^+^ + K^+^)-ATPase activity sensitive to HG-9-91-01. Ouabain-sensitive ATPase activity was measured in the homogenate obtained from the kidney cortex in the absence (open bars; total (Na^+^ + K^+^)-ATPase activity) and in the presence of 1 µM HG-9-91-01 (hachured bars; (Na^+^ + K^+^)-ATPase activity resistant to SIK inhibition). The difference between total (Na^+^ + K^+^)-ATPase activity and (Na^+^ + K^+^)-ATPase activity resistant to SIK inhibition is the (Na^+^ + K^+^)-ATPase activity sensitive to SIK inhibition and is expressed in the table below the graph. Values are expressed as a percentage of control. Different superscript lowercase letters indicate statistically significant differences (*p* < 0.05). One-way ANOVA followed by Sidak’s post hoc test was applied to detect differences in the effects of HSD (^a^—CTRL vs. CTRL-Salt); HTN (^b^—CTRL vs. DOCA); the combination of HSD and HTN (^c^—CTRL vs. DOCA-Salt); HSD-exacerbated HTN (^d^—DOCA vs. DOCA-Salt); and HTN-exacerbated HSD (^e^—CRTL-Salt vs. DOCA-Salt). Different superscripted lowercase letters represented in the Figure only in case of statistically significant differences (*p* < 0.05).

**Figure 10 ijms-24-07887-f010:**
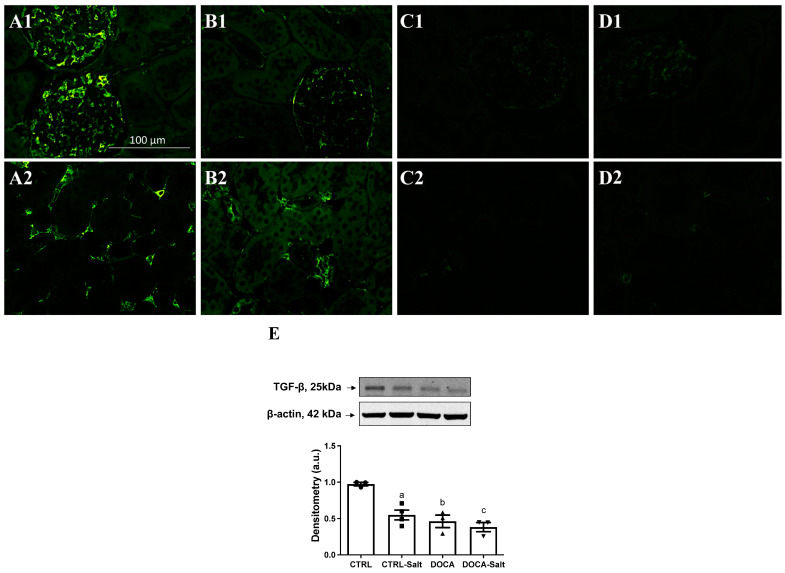
The impact of high-salt diet (HSD) in combination or not with hypertension (HTN) on TGF-β1. Immunofluorescence was performed as described in Materials and Methods. Paraffin-embedded kidney poles from CTRL (**A1**,**A2**), CTRL-Salt (**B1**,**B2**), DOCA (**C1**,**C2**), and DOCA-Salt (**D1**,**D2**) were incubated overnight with anti-rabbit TGF-β1 antibody (1:200) at 4 °C, followed by incubation with specific secondary antibody (1:5000, Alexa Fluor 488-labeled secondary antibody, Invitrogen, Waltham, MA, USA). Representative photomicrographs (×400) of kidney poles in glomeruli (upper panels) and tubulointerstitial areas (lower panels). (**E**) TGF-β1 protein content in kidney cortexes. Upper panel: representative images of TGF-β1 detection using mouse monoclonal anti-TGF-β1 (1:200; Santa Cruz Biotechnology, Dallas, TX, USA) and polyclonal β-actin (1:5000, Santa Cruz Biotechnology, Dallas, TX, USA). Lower panel: densitometric analysis of the immunoreactive band (25 kDa) correlated by the β-actin expression. The data are presented as the mean ± SEM. Different superscript lowercase letters indicate statistically significant differences (*p* < 0.05). One-way ANOVA followed by Sidak’s post hoc test was applied to detect differences in the effects of HSD (^a^—CTRL vs. CTRL-Salt); HTN (^b^—CTRL vs. DOCA); the combination of HSD and HTN (^c^—CTRL vs. DOCA-Salt); HSD-exacerbated HTN (^d^—DOCA vs. DOCA-Salt); and HTN-exacerbated HSD (^e^—CRTL-Salt vs. DOCA-Salt). Different superscripted lowercase letters represented in the figure only in case of statistically significant differences (*p* < 0.05).

**Figure 11 ijms-24-07887-f011:**
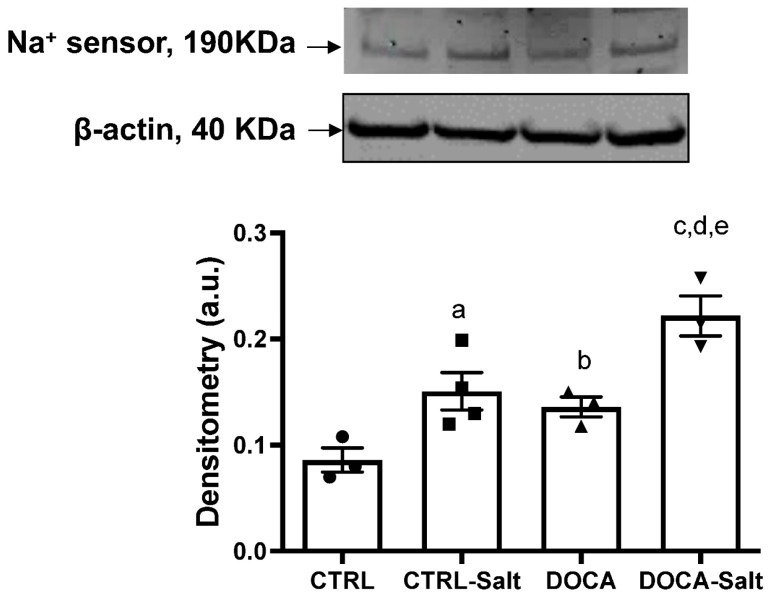
Effect of high-salt diet (HSD) in combination or not with hypertension (HTN) on Nax content and activity. Wistar rats were subjected to (i) normal salt diet (NSD; CTRL), (ii) HSD (CTRL-Salt), (iii) NSD and treated with deoxycorticosterone acetate (DOCA), and (iv) HSD treated with DOCA (DOCA-Salt), as described in Materials and Methods. Upper panel: representative images of Nax detection using Nax rabbit Scn7a polyclonal antibody (1:500; Abcam, Cambridge, UK) and polyclonal β-actin (1:5000, Santa Cruz Biotechnology, Dallas, TX, USA). Lower panel: densitometric analysis of the immunoreactive band (190 kDa) correlated by the β-actin expression. The data are presented as the mean ± SEM. Different superscript lowercase letters indicate statistically significant differences (*p* < 0.05). One-way ANOVA followed by Sidak’s post hoc test was applied to detect differences in the effects of HSD (^a^—CTRL vs. CTRL-Salt); HTN (^b^—CTRL vs. DOCA); the combination of HSD and HTN (^c^—CTRL vs. DOCA-Salt); HSD-exacerbated HTN (^d^—DOCA vs. DOCA-Salt); and HTN-exacerbated HSD (^e^—CRTL-Salt vs. DOCA-Salt). Different superscripted lowercase letters represented in the Figure only in case of statistically significant differences (*p* < 0.05).

**Figure 12 ijms-24-07887-f012:**
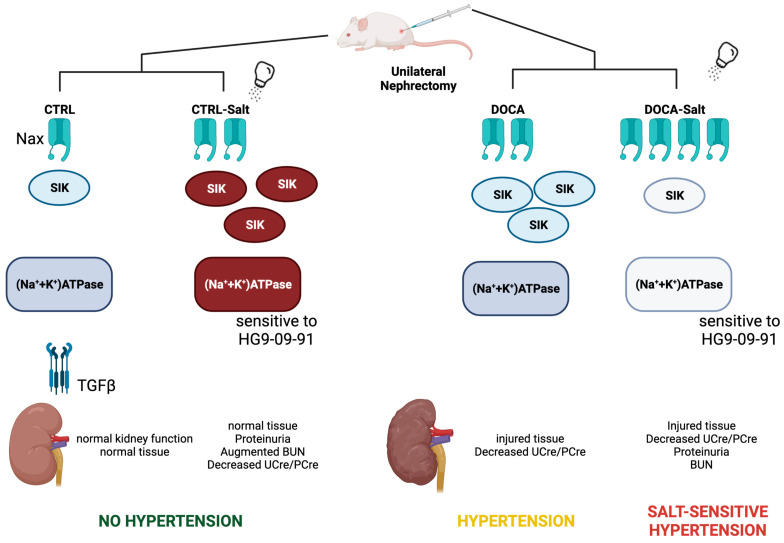
The triad Nax/SIK/(Na^+^ + K^+^)-ATPase is regulated by both high-salt diet (HSD) and hypertension (HTN), presenting a dual effect. Uninephrectomized Wistar rats subjected to HSD is a salt-resistant model because it does not develop HTN. The unique effect of Na^+^ on the kidney involves overexpression and overstimulation of the triad, marked in red. Normal kidney tissue, proteinuria, and a decreased U_Cre_/P_Cre_ ratio are compatible with the initial phase of chronic kidney disease. The pressure component also over expresses the triad, but SIK and (Na^+^ + K^+^)-ATPase activities are close to control levels, marked in blue. Downregulation of the triad is evident in the combination of HSD and HTN. Although DOCA-Salt rats presented the highest levels of Nax, their activity and expression levels of SIK and (Na^+^ + K^+^)-ATPase were close to that of CTRL rats (light blue). The association of HSD with DOCA promotes salt-sensitive HTN, which speeds up the impairment of kidney function, observed by decay in renal parameters and renal structure, establishing chronic kidney disease. TGF-β expression is diminished in all three experimental conditions, leading to the hypothesis that increased Nax expression is associated with kidney fibrosis. Created with BioRender.com HU24KVSDQ6.

**Table 1 ijms-24-07887-t001:** Effects of HSD and HTN on general physiological parameters.

Parameters	CTRL (*n* = 5)	CTRL-Salt (*n* = 5)	DOCA (*n* = 5)	DOCA-Salt (*n* = 5)
Body mass (g)	305 ± 14	315 ± 0.5 (*p* = 0.9873)	326 ± 21 (*p* = 0.7630)	346 ± 4 (*p* = 0.1286; *p* = 0.7972; *p* = 0.3822)
Kidney mass (g)	1.5 ± 0.1	1.9 ± 0.1 (*p* = 0.0320)	2 ± 0.1 ^b^ (*p* = 0.0041)	2.4 ± 0.1 ^c,e^ (*p* < 0.0001; *p* = 0.0320; *p* = 0.0041)
Kidney index (mg/g)	4.8 ± 0.1	5.8 ± 0.2 (*p* = 0.0893)	6.2 ± 0.2 (*p* = 0.0063)	7.0 ± 0.5 ^c,e^ (*p* < 0.0001; *p* = 0.2558; *p* = 0.0256)
SBP (mmHg)	104 ± 4	112 ± 1.2 (*p* = 0.8959)	145 ± 3 ^b^ (*p* < 0.0001)	160 ± 4 ^c,d,e^ (*p* < 0.0001; *p* = 0.0136; *p* < 0.0001)
Heart rate (bpm)	280 ± 7	337 ± 10 ^a^ (*p* = 0.0008)	452 ± 4 ^b^ (*p* < 0.0001)	463 ± 5 ^c,e^ (*p* < 0.0001; *p* = 0.2385; *p* < 0.0001)
Water intake (mL/24 h)	26 ± 2	49 ± 5 ^a^ (*p* = 0.0173)	24 ± 2 (*p* = 0.9993)	55 ± 6 ^c,d^ (*p* = 0.0007; *p* < 0.0001; *p* = 0.8677)
Urine volume (mL/24 h)	9 ± 3	29 ± 3 ^a^ (*p* = 0.0022)	7 ± 1 (*p* = 0.9964)	36 ± 5 ^c,d^ (*p* < 0.0001; *p* < 0.0001; *p* = 0.4246)

Twenty 8-week-old male Wistar rats underwent a unilateral nephrectomy and were randomly divided into 4 groups after recovering from surgery, as described in Materials and Methods [24]: (i) normal salt diet (NSD; CTRL), (ii) HSD (CTRL-Salt), (iii) NSD and treated with deoxycorticosterone acetate (DOCA), and (iv) HSD treated with DOCA (DOCA-Salt). On day 28, the following parameters were measured: weight, heart rate, blood pressure, and 24-h water intake and urine excretion from individual metabolic cages. The data are presented as the mean ± SEM. Different superscript lowercase letters indicate statistically significant differences (*p* < 0.05). One-way ANOVA followed by Sidak’s post hoc test was applied to detect differences in the effects of HSD (^a^—CTRL vs. CTRL-Salt); HTN (^b^—CTRL vs. DOCA); the combination of HSD and HTN (^c^—CTRL vs. DOCA-Salt); HSD-exacerbated HTN (^d^—DOCA vs. DOCA-Salt); and HTN-exacerbated HSD (^e^—CRTL-Salt vs. DOCA-Salt). Different superscripted lowercase letters represented in the Table only in case of statistically significant differences (*p* < 0.05). SBP: systolic blood pressure.

**Table 2 ijms-24-07887-t002:** The impact of HSD and HTN on kidney function in Wistar rats.

Parameters	CTRL (*n* = 5)	CTRL-Salt (*n* = 5)	DOCA (*n* = 5)	DOCA-Salt (*n* = 5)
BUN (mg/dL)	33 ± 1.2	43 ± 2.4 ^a^ (*p* = 0.0068)	31 ± 1.6 (*p* = 0.8622)	39 ± 1.5 ^c,d^ (*p* = 0.0868; *p* = 0.0114; *p* = 0.6234)
Proteinuria (mg/24 h)	6.6 ± 1.3	21.5 ± 3.9 ^a^ (*p* = 0.0007)	8.2 ± 0.7 (*p* = 0.9866)	16 ± 1.7 ^c,d^ (*p* = 0.0243; *p* = 0.0497; *p* = 0.3452)
P_Cre_ (mg/dL)	0.46 ± 0.03	0.49 ± 0.04 (*p* = 0.9971)	0.56 ± 0.07 (*p* = 0.7118)	0.75 ± 0.05 ^c,d,e^ (*p* = 0.0043; *p* = 0.0439; *p* = 0.0136)
U_Cre_ (mg/dL)	178 ± 34	33 ± 4.6 ^a^ (*p* < 0.0001)	128 ± 12 (*p* = 0.2449)	35 ± 3.2 ^c,d^ (*p* < 0.0001; *p* = 0.0037; *p* > 0.9999)
U_Cre_/P_Cre_ ratio	402 ± 93	70 ± 14 ^a^ (*p* = 0.0007)	207 ± 30 ^b^ (*p* = 0.0410)	46 ± 4 ^c,d^ (*p* = 0.0002; *p* = 0.0116; *p* = 0.9981)
P_Na_ (mEq/L)	140 ± 4	155 ± 6 (*p* = 0.9407)	153 ± 12 (*p* = 0.9415)	131 ± 9 (*p* = 0.9787; *p* = 0.3905; *p* = 0.5485)
U_Na_ (mEq/L)	172 ± 0.9	476 ± 12 ^a^ (*p* < 0.0001)	161 ± 17 (*p* = 0.9983)	399 ± 26 ^c,d^ (*p* < 0.0001; *p* < 0.0001; *p* = 0.1852)
FL_Na_ (mEq/min)	0.17 ± 0.005	0.20 ± 0.008 (*p* = 0.4572)	0.18 ± 0.014 (*p* = 0.9139)	0.11 ± 0.008 ^c,d,e^ (*p* < 0.0243; *p* = 0.0003; *p* = 0.0004)
FE_Na_ (%)	0.63 ± 0.14	4.8 ± 0.3 ^a^ (*p* = 0.0357)	0.47 ± 0.08 (*p* > 0.9999)	8.7 ± 1.07 ^c,d,e^ (*p* < 0.0001; *p* < 0.0001 *p* = 0.0351)

Twenty 8-week-old male Wistar rats underwent a unilateral nephrectomy and were randomly divided into 4 groups after recovering from surgery, as described in Materials and Methods [24]: (i) normal salt diet (NSD; CTRL), (ii) HSD (CTRL-Salt), (iii) NSD and treated with deoxycorticosterone acetate (DOCA), and (iv) HSD treated with DOCA (DOCA-Salt). On day 28, rats were weighed and individually housed in metabolic cages for 24 h urine collections and 24 h water intake to evaluate renal function parameters. The data are presented as the mean ± SEM. Different superscripted lowercase letters indicate statistically significant differences (*p* < 0.05). One-way ANOVA followed by Sidak’s post hoc test was applied to detect differences in the effects of HSD (^a^—CTRL vs. CTRL-Salt); HTN (^b^—CTRL vs. DOCA); the combination of HSD and HTN (^c^—CTRL vs. DOCA-Salt); HSD-exacerbated HTN (^d^—DOCA vs. DOCA-Salt); and HTN-exacerbated HSD (^e^—CRTL-Salt vs. DOCA-Salt). Different superscripted lowercase letters represented in the Table only in case of statistically significant differences (*p* < 0.05). BUN: blood urea nitrogen; P_Cre_: plasma creatinine; U_Cre_: urinary creatinine; U_Cre_/P_Cre_ ratio: the ratio between urinary creatinine and plasma creatinine; P_Na_: plasma Na^+^ concentration; U_Na_: urinary Na^+^ concentration; FL_Na_: filtered load Na^+^; and FE_Na_: fractional excretion of Na^+^ (the percentage of FL_Na_ that was excreted in the urine).

**Table 3 ijms-24-07887-t003:** Immunolocalization of SIK isoforms along the nephron.

SIK Isoform	Glomerulus	Proximal Tubule	Thick Ascending Limb	Collecting Duct
SIK1	Mesangial cells	−	+	+
SIK2	+	+	+	+
SIK3	Mesangial cells	Brush border	−	Principal cells

−: negative expression. +: positive expression.

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
