# Peer review of "The Triad Na+ Activated Na+ Channel (Nax)—Salt Inducible KINASE (SIK) and (Na+ + K+)-ATPase: Targeting the Villains to Treat Salt Resistant and Sensitive Hypertension"

_ijms, 2023, doi:10.3390/ijms24097887_

Round 1

Reviewer 1 Report

The results presented are quite interesting and well summarized in Figure 12 of the manuscript. I think they deserve to be published, but I would suggest that the Author consider a few remarks to improve their paper (please see the attached file).

Author Response

IJMS-2230446

MS Title: The triad Na+ activated Na+ channel (Nax) - Salt Inducible Kinase (SIK) and (Na++K+)-ATPase: targeting the villains to treat salt resistant and sensitive hypertension.

Answers to the Reviewers

            We appreciate the detailed critique and the thought-provoking issues that have been raised. We have carefully considered all the comments and revised the paper point-by-point as appropriate. Our detailed responses are as follows (modifications were made in red in the marked version).

REVIEWER #1

  1. Citation: “The Abstract may need to be thoroughly revised: in my opinion, there is not a clear order in what is reported and the main findings of the study are not appropriately mentioned.

For example:

  • l.35: The syntax is not clear (maybe a verb is needed)
  • l.38: not entirely clear what “±” in “±200 g” means
  • l.40: if possible, define “DOCA” before using this abbreviation
  • l.49: were detected… in which animal model?

Answer: We appreciate the critiques in the Abstract and apologize for not appropriately mentioning the main findings of the study. We have made a complete change in the abstract, especially regarding the sentence in l.50: “HSD itself attenuates the response of (Na++K+)-ATPase activity to SIK”. We changed for: “The study showed that the response of (Na++K+)-ATPase activity to SIK was attenuated in rats subjected to HSD, independent of HTN…” (l.55).

  1. Citation: “l.50: “HSD itself attenuates the response of (Na+K+)-ATPase activity to SIK”: how is that justified by the data presented in the Results? Is this conclusion drawn from the effect of SIK inhibitors on (Na+K+)-ATPase activity?

It may be worth considering ending the Abstract with a concise, however general, statement that remarks, for example, that the Authors present a study on the effects of acquired causes of kidney damage on the Nax/SIK/(Na+K+)-ATPase signal transduction pathway.

Answer: We have changed Figure 9C for better comprehension. As demonstrated, in CTRL rats, the (Na++K+)ATPase activity in the presence of  HG 9-91-01 represents, in turn, 50 % of the total (Na++K+)ATPase activity. In this way, we can infer that in CTRL rats (Na++K+) ATPase activity is sensitive to SIK inhibition: once if SIK is inhibited, (Na++K+)ATPase activity is also inhibited. HSD augmented in 73% the fraction of (Na++K+)-ATPase activity sensitive to HG-9-91-01 compared to CTRL. In DOCA rats, the (Na++K+)-ATPase activity sensitive to SIK inhibition was almost inexistent while in the DOCA-Salt, the SIK sensitivity was restored to CTRL levels. This leads us to conclude that HSD turns (Na++K+)ATPase activity sensitive to SIK inhibition. We have included this interpretation in l. 268-l.280, l.342-l.355 and l.570-l.583.

  1. Citation: “In fact, the first paragraph in the Introduction may find a better collocation in the Discussion section.

Answer: We moved this paragraph to the Discussion section (l.414).

  1. Citation: “In general, I suggest that the authors ask an English-proficient colleague to revise the manuscript to improve understandability, in particular in the Abstract and Introduction section.

Answer: The revised version of the manuscript was edited by a native English speaker.

  1. Citation: “l.176:As a result, kidney function is jeopardized by HSD and HTN combination” → the authors may need a comparison between CTRL-Salt and DOCA-Salt groups to justify this (reasonable) statement.

Answer: We think that the sentence: “Plasma Na+ concentration did not vary in any condition.” got in the way between the comparison with CTRL-Salt and DOCA-Salt. We modified the sentence position (now l.179) to make clear the justification of the statement pointed by the Reviewer.

  1. Citation: “l.298: “The concomitant HSD intake downregulated both SIK and (Na+K+)-ATPase activity leading to the higher level of FENa.” Why? This statement does not seem entirely justified by what has been claimed before. Instead, it may be observed that HSD alone was shown to increase SIK and (Na+K+)-ATPase activity (Figures 8 and 9), which is rather the opposite.

Answer: For a better understanding we rephrased the sentence to:

l.371- l.320: “Our study showed reduced SIK protein content which impacted (Na++K+)-ATPase activity; we attributed this impact to the increased levels of SBP by HSD and HTN. The decreased (Na++K+)-ATPase activity may explain, in part, the higher level of FENa.”

We wanted to make clear that in the DOCA-Salt rat (where HTN is combined with HSD) the downregulation of the SIK and (Na++K+)-ATPase activities might contribute to the augmented FENa. It is well established that (Na++K+)-ATPase is responsible to tubular Na+ reabsorption. In this way, when (Na++K+)-ATPase activity is diminished, Na+ excretion is augmented.

  1. Citation: “l.320: sik1 should be capitalized, as it is a human gene.

Answer: Correction was made.

  1. Citation: “ll.326-327: this first part of the statement is not entirely clear.

Answer: We intended to explain the relationship between SIK1, (Na++K+)-ATPase activity and hypertension. To clarify this issue, we completed the statement with more information. We compared what is demonstrated in the vasculature and in the kidney. In vascular smooth muscle cells, the augmented kinase activity of SIK1, leads to higher (Na++K+)-ATPase activity, and lower levels of intracellular Ca2+, which leads to vascular relaxation. In the DOCA-Salt, we observed the opposite: lower SIK1 content, decreased (Na++K+)-ATPase activity, and the lower response of the (Na++K+)-ATPase activity to SIK inhibition. This was associated with an unresolved adaptive response to control Na+ reabsorption, but it causes kidney injury and salt-sensitive hypertension (l.342 – l.355).

  1. Citation: “l.338: “collecting” duct

Answer: We don’t understand the correction suggested since collecting seems to be spelling correctly.

  1. Citation: “ll.374-376: “We hypothesize that Nax does not respond to the pressure component which increases kidney fibrosis, increasing fibronectin deposition, which could explain the renal injury”; yet, Figure 11 shows that Nax expression is higher in DOCA rats than in controls. How would the authors justify this finding?

Answer: This sentence is not correct. We changed for l.408 “In DOCA-Salt rat, there is an intense Nax expression in the renal cortex that may contribute to an intense fibronectin deposition, which could explain the renal injury. Indeed, exacerbation of homeostatic Na+ regulation via Nax proceeds to induce pathological tissue phenotypes via promotion of pro-inflammatory and pro-fibrotic responses.”

  1. Citation: “ll. 384-385: “The presence of the pressure component negatively regulates SIK activity, which promotes a dysregulation of (Na+K+)-ATPase”; again, a direct comparison between CTRL-Salt and DOCA-Salt models (in SIK activity at least) would help justifying this reasonable statment.

Answer: We removed this sentence and explain this issue better in the item 8.

  1. Citation: “ll.392-393: “HSD itself activates the triad Nax/SIK/(Na++K+)-ATPase, but it does not provoke HTN, although it initiates the progression of chronic kidneydisease.”; this does not look ike the focus of the study, though. The authors caused hypertension on purpose with DOCA, they did not look into whether HSD causes hypertension.

Answer: We agreed with the Reviewer and removed this sentence.

  1. Citation: “l.395: salt-sensitive HTN was associated with kidney fibrosis and with Nax overexpression. A direct association between Nax overexpression and kidney fibrosis, other than being both caused by salt-sensitive HTN, does not appear entirely justified. This should be made a little more clear in the final statement.

Answer: We clarified the statement adding the information that intense homeostatic Na+ regulations via Nax provoke an intense tissue pro-inflammatory and pro-fibrotic status. This led us to hypothesize that the fibrosis encountered in the DOCA-Salt rat is the consequence of an intense Nax expression (l.408 – l.413).

  1. Citation: “Figures 3 to 6: the caption should report that the presence of “(D) anti-rataquaporin-

2” stands for the collecting duct.

Answer: The correction was made.

  1. Citation: “Figures 6 does not show the proximal tubule, which should be reported as a negative control.

Answer: We included new images of immunodetection using SIK antibody and aquaporin 1 (APQ1), a marker of the proximal tubule. In all experimental groups (CTRL, CTRL-Salt, DOCA, and DOCA-Salt), SIK1 does not colocalize in the proximal tubule (Figure 6B).

  1. Citation: “The text mentions figure 7A, but only one histogram is reported in Figure 7.

Answer: The correction was made.

  1. Citation: “I find figure 8 a little too difficult to understand. Also, histograms in the insets are too small and do not have hachured bars, contrary to what is mentioned in the caption (it looks as if the main histograms and inset histograms were switched). What do the bars in the inset histograms represent? Inhibited or unhinibited SIK activity? From the numbers in the main text, it seems like it is the inhibited case.

Anyway, the correspondence between the histograms in each panel, and between the panels and what is reported in the text, is not entirely clear. The findings seem interesting and may be considered the core results of the study, but I would like to see them more clearly.

Answer: The Figure and the corresponding caption were modified for better understanding. The rationale to determine a kinase activity is based on the ability of the enzyme to transfer the phosphate from the [γ-P]ATP into the histone. To specifically determine SIK activity, we measured phospho-histone formation in the absence (total phospho-histone formation) and presence of two SIK different inhibitors (phospho-histone formation resistant to SIK inhibition). The SIK inhibitors used was MRT 199665 (Figure 8A) and HG-9-91-01 (Figure 8B). The difference between total phospho-histone formation and the phospho-histone formation resistant to SIK inhibition represents the SIK activity and is expressed in the table above the Figure 8A (SIK activity sensitive to MRT 199665) and Figure 8B (SIK activity sensitive to HG-9-91-01).

  1. Citation: “Figure 11: from reading the caption, I understand that at least another colored histogram (with the activity of SIK and (Na+K+)-ATPase) is missing.

Answer: We apologize for the mistake. The caption of Figure 11 was missing, and the caption of Figure 12 was inappropriate placed in Figure 11 caption. The correction was made.

  1. Citation: “Figure 12 needs a caption, however brief; otherwise, I find it quite good at recapitulating the main findings of the study.”

Answer: We apologize for the mistake. The caption of Figure 11 was missing and the caption of Figure 12 was inappropriately placed in the Figure 11 caption. The correction was made, according to the Reviewer’s suggestion.

  1. Citation: “I also suggest that the Author add a few final remarks, possibly based on what is reported

in Figure 12, which may help summarizing the most interesting results of their study.

Answer: Concluding remarks were added at the end of the Discussion section.

Reviewer 2 Report

The authors have provided a excellently driven experimental study. The manuscript is nicely written and sound from a maethodological point of view. Please mind the following comments:

1. Please add the name of the first author (author et al.) in the material and method section, when referring to previously published work (ex: lines 407,420).

2. Please add legends to the tables where you name the acronyms used among the tables.

3. Please add p values within the tables.

4. Please correct minor English language mistakes

Author Response

IJMS-2230446

MS Title: The triad Na+ activated Na+ channel (Nax) - Salt Inducible Kinase (SIK) and (Na++K+)-ATPase: targeting the villains to treat salt resistant and sensitive hypertension.

Answers to the Reviewers

            We appreciate the detailed critique and the thought-provoking issues that have been raised. We have carefully considered all the comments and revised the paper point-by-point as appropriate. Our detailed responses are as follows (modifications were made in red in the marked version).

REVIEWER #2

  1. Citation: “Please add the name of the first author (author et al.) in the material and method section, when referring to previously published work (ex: lines 407,420).

Answer: The correction was made.

  1. Citation: “Please add legends to the tables where you name the acronyms used among the tables.

Answer: The correction was made.

  1. Citation: “Please add p values within the tables.

Answer: We have added the p values within the tables.

  1. Citation: “Please correct minor English language mistakes

Answer: The revised version of the manuscript was edited by a native English speaker.

Round 2

Reviewer 1 Report

Please consider the following points:

-2. -> according to your explanation, I would rather conclude that HSD alone (that is: without HTN, not independent of HTN) exacerbates, increases (rather than attenuates) the percentage of SIK-sensitive ATP activity. Is that correct? otherwise, I kindly ask you to show me where my reasoning is wrong.

-5. -> I still could not see any p value related to the comparison between CTRL-salt and DOCA-salt: no one among a), b), c), and d) related to this particular comparison. It should not be difficult to find it though, since the post hoc pairwise comparisons should automatically include CTRL-salt and DOCA-salt too.

-6.-> Figure 7 shows an increased SIK protein content in the experimental conditions, compared to CTRL. Again, I beg the Authors to show me how I am misinterpreting these results.

-11. -> see point 5.

-lines  485-486 in the manuscript -> "decreased (Na++K+ 485 )-ATPase activity led to a diminished Na+ reabsorption"; this is a reasonable hypothesis, but still a hypothesis; a modal verb or adverbs such as "likely" would be preferred then. Also, SIK1 levels in DOCA-salt rats are comparable to SIK1 levels in CTRL, as is APTase activity. How would that be an explanation for a pathogenic mechanism in DOCA-salt rats? Perhaps the Author should mention the fact that Nax levels differed between the two groups.

-concluding remark 2,3, and 4 -> which rats??? since two different populations are discussed each time, it must be mentioned which of the two exhibits more or less something than the other. Additionally, CTRL-salt vs DOCA-salt are still not mentioned.

-concluding remark 5 -> how is that justified?? is there a control group of CTRL-salt rats in which SIK1 and Nax contents were restored to normal, and in which TGFbeta1 levels were dosed?

-concluding remark 7 (it also needs rephrasing) -> "salt-sensitive HTN rats, SIK, and (Na++K+)-ATPase are downregulated" -- compared to which groups?? not CTRL I think -- "but not Nax" -- this should be highlighted. In other words, DOCA-salt rats appear to have a diminished response in SIK1 content and activitity, and ATPase activity compared to the DOCA and HSD rats, as the levels of those variables in DOCA-salt rats were comparable to CTRL (might that be a manifestation of an "exhaustion" in adjustment mechanisms?). Nax levels, instead, were highest in DOCA-salt rats, which could explain the fibrogenesis etc.

I look forward to the Authors feedback and apologize in advance if my doubts are less than justified. However, at present I still believe that this paper needs some additional revision.

Author Response

IJMS-2230446

2nd Round

MS Title: The triad Na+ activated Na+ channel (Nax) - Salt Inducible Kinase (SIK) and (Na++K+)-ATPase: targeting the villains to treat salt resistant and sensitive hypertension.

Answers to the Reviewer 1 (R1):

Many thanks to R1 for the suggestions.  We have carefully considered all the comments and revised the paper point-by-point as appropriate. Our detailed responses are as follows (modifications were made in red in the marked version).

Remining concerns of R1:

  1. According to your explanation, I would rather conclude that HSD alone (that is: without HTN, not independent of HTN) exacerbates, increases (rather than attenuates) the percentage of SIK-sensitive ATP activity. Is that correct? otherwise, I kindly ask you to show me where my reasoning is wrong.

Answer: R1 is correct. The conclusion was re-worded and clearly stated in the Discussion (lines 321-324; lines 347-352).

  1. I still could not see any p value related to the comparison between CTRL-salt and DOCA-salt: no one among a), b), c), and d) related to this particular comparison. It should not be difficult to find it though, since the post hoc pairwise comparisons should automatically include CTRL-salt and DOCA-salt too.

Answer: We apologize for mistaking the comparisons. We have corrected all comparisons accordingly.

  1. Figure 7 shows an increased SIK protein content in the experimental conditions, compared to CTRL. Again, I beg the Authors to show me how I am misinterpreting these results.

Answer: We used One-way ANOVA followed by Sidak-s post hoc test to detect the differences of the effecst in the following experimental groups:

  • HSD: we compared CTRL vs CTRL-Salt – the p value was <0.0001 and was indicated by letter “a” in the graph.
  • HTN: we compared CTRL vs DOCA – the p value was 0.0141 and was indicated by letter “b” in the graph.
  • the combination of HSD and HTN: we compared CTRL vs DOCA-Salt – the p value was 0.9994; because it is not statistically different, the letter “c” was not represented.
  • HSD-exacerbated HTN: we compared DOCA vs DOCA-Salt – the p value was 0.0231 and was indicated by letter “d” in the graph.
  • HTN-exacerbated HSD: we compared CTRL-Salt vs DOCA-Salt – the p value was <0.0001 and was indicated by letter “e” in the graph.

The p values indicate that SIK1 protein content was augmented in CTRL-Salt and DOCA in comparison to CTRL. The SIK1 protein content in the DOCA-Salt was similar to CTRL and diminished in comparison to DOCA. Therefore, HSD alone or HTN alone augment SIK1 protein content in the kidney cortex, while the combination does not. SIK1 protein content in the DOCA-Salt is also diminished in comparison to CTRL-Salt. This information was clarified in the Result section, lines 240-244.

  1. see point 5.

Answer: Thanks for the suggestion. We have included CTRL-Salt and DOCA-Salt in the comparison as suggested.

English revision was done by a native English speaker (Vallotton, Z.).

New concerns of R1:

  1. A) Lines 485-486 in the manuscript -> "decreased (Na++K+)-ATPase activity led to a diminished Na+ reabsorption"; this is a reasonable hypothesis, but still a hypothesis; a modal verb or adverbs such as "likely" would be preferred then.

Answer: You are right. The sentence was re-worded (line 352).

  1. B) Also, SIK1 levels in DOCA-salt rats are comparable to SIK1 levels in CTRL, as is APTase activity. How would that be an explanation for a pathogenic mechanism in DOCA-salt rats? Perhaps the Author should mention the fact that Nax levels differed between the two groups.

Answer: That is a good point and we have included in the manuscript (lines: 321-325) and in the Concluding remarks (# 4, lines 458-462).

  1. C) concluding remark 2,3, and 4 -> which rats??? since two different populations are discussed each time, it must be mentioned which of the two exhibits more or less something than the other. Additionally, CTRL-salt vs DOCA-salt are still not mentioned.

Answer: We have included the missing information.

  1. D) concluding remark 5 -> how is that justified?? is there a control group of CTRL-salt rats in which SIK1 and Nax contents were restored to normal, and in which TGFbeta1 levels were dosed?

Answer: At this moment, we do not have a good explanation for the TGFbeta1 findings. We will investigate further these aspects and we discussed it in lines: 406-429)No conclusion remark was raised on this issue.

  1. E) concluding remark 7 (it also needs rephrasing) -> "salt-sensitive HTN rats, SIK, and (Na++K+)-ATPase are downregulated" -- compared to which groups?? not CTRL I think -- "but not Nax" -- this should be highlighted. In other words, DOCA-salt rats appear to have a diminished response in SIK1 content and activitity, and ATPase activity compared to the DOCA and HSD rats, as the levels of those variables in DOCA-salt rats were comparable to CTRL (might that be a manifestation of an "exhaustion" in adjustment mechanisms?). Nax levels, instead, were highest in DOCA-salt rats, which could explain the fibrogenesis etc.

Answer: Concluding remark # 6 (in the old version, concluding remark 7) was reworded accordingly
